# Sub-millisecond lithiothermal synthesis of graphitic meso–microporous carbon

Huimin Zhang[1], Jingyi Qiu[1], Jie Pang[2], Gaoping Cao[1], Bingsen Zhang [3], Li Wang [4], Xiangming He [4], Xuning Feng [4], Shizhou Ma[1], Xinggao Zhang[1], Hai Ming[1], Zhuangnan Li[5], Feng Li [3] ✉ & Hao Zhang [1] ✉

Porous carbons with concurrently high specific surface area and electronic conductivity are desirable by virtue of their desirable electron and ion transport ability, but conventional preparing methods suffer from either low yield or inferior quality carbons. Here we developed a lithiothermal approach to bottom–up synthesize highly meso–microporous graphitized carbon (MGC). The preparation can be finished in a few milliseconds by the self-propagating reaction between polytetrafluoroethylene powder and molten lithium (Li) metal, during which instant ultra-high temperature (>3000 K) was produced. This instantaneous carbon vaporization and condensation at ultra-high temperatures and in ultra-short duration enable the MGC to show a highly graphitized and continuously cross-coupled open pore structure. MGC displays superior electrochemical capacitor performance of exceptional power capability and ultralong-term cyclability. The processes used to make this carbon are readily scalable to industrial levels.

Highly graphitized meso–microporous carbon materials have generated a great deal of technological interest due to both high specific surface area (SSA) and electronic conductivity[1,2]. By virtue of their ability to overcome the charge/mass-transport limitations encountered in traditional activated carbon, meso–microporous graphitized carbon (MGC) plays a significant role in the area of electrochemical capacitor[1,3], lithium-sulfur batteries[4], catalysis support[5], and separation and storage, as capacitive deionization (CDI), etc.[6,7]. The majority of the porous carbons are prepared through thermal strategies, either bottom–up methods like chemical vapor deposition (CVD)[8,9] and arc-discharge evaporation involving carbon vaporization and fast condensation, or top–down ways like pyrolysis, carbonization, and graphitization, both of which undertake a series of carbon bond breaking and reforming processes[10,11].

To date, ultrahigh temperature strategies like the bottom–up method (such as CVD)[12,13] and graphitization over 2000 °C can induce the formation of $sp^2$-bonded carbon, producing carbon nanotubes[14], fullerenes[15], or onion-like carbons with excellent electronic conductivity[1]. However, the former strategy suffers from relatively low yield and higher cost, while the long-term treatment of the latter method leads to severe loss of pore structures. Moderate temperature strategies, such as pyrolysis or chemical activation under 1000 °C, can produce porous carbons with high SSA, while they face the dilemma of low temperature going against the formation of graphitized structures (Supplementary Figs. 1 and 2), and thus leading to low electronic conductivity[16]. Laser scribing is a strategy based on the photothermal and photochemical reactions between carbon precursors and laser spot, which opens a pathway for graphene-based electrode materials with high conductivity[17]. While the limited SSA of laser scribing graphene limits its application as electrodes for supercapacitors. It is found that the halogen elements (F, Cl) connecting to vinylidene units are highly reactive so that dehalogenation can take place at room temperature in the presence of strong inorganic alkaline. Dehalogenation-based carbonization based on halogenated polymers, such as polyvinyl fluoride (PVDF), and polyvinylidene chloride, can be used to prepare high SSA porous carbons[18,19]. However, the pore size

[1]Beijing Key Laboratory of Advanced Chemical Energy Storage Technologies and Materials, Research Institute of Chemical Defense, Beijing, China. [2]School of Energy Science and Technology, Henan University, Zhengzhou, China. [3]Shenyang National Laboratory for Materials Science, Institute of Metal Research, Chinese Academy of Sciences, Shenyang, China. [4]Institute of Nuclear and New Energy Technology, Tsinghua University, Beijing, China. [5]Department of Material Science and Metallurgy, University of Cambridge, Cambridge, UK. ✉e-mail: fli@imr.ac.cn; dr.h.zhang@hotmail.com

distribution for the porous carbon obtained is relatively wider, and graphitization needs to be improved. Therefore, a methodology is desired for the development of porous carbons with both high surface area and developed graphitized structure.

We report the synthesis of MGC by intensive and ultrafast exothermic reaction between polytetrafluoroethylene (PTFE) powder and molten Li metal, named lithiothermal method, which is self-propagating high-temperature synthesis. This reaction can be initiated by direct immersion of PTFE powder into the molten Li with a lateral shear (Fig. 1 and Supplementary Fig. 3). The heat released from the strongly exothermic reaction creates a combustion wave and spontaneously propagates to the remaining reactants in self-sustaining behavior. The resultant MGC shows a high surface area of over 2000 m²/g and good graphitized crystallinity, due to the instantaneous carbon vaporization and condensation at the high temperature of reaction (>3000 K, Supplementary Note 1 and Fig. 4).

## Results

The preparation of porous graphitized carbon is schematically described in Fig. 1A. Under Ar atmosphere, initial thermal stimulus with a lateral shear[20] (200 °C, leading to the melting of lithium foil with a melting point of 180 °C) is applied to lithium foil wrapped PTFE powder reactants (Supplementary Fig. 3). Subsequently, the mixture fiercely detonated, producing graphitized carbon granules containing LiF (Supplementary Fig. 5) immediately, during which the high enthalpy of Li−F bond formation serves as the thermodynamic driving force to sustain the reaction process ($\Delta_f H_{298}^{\theta}$(LiF) = −616.9 kJ/mol). The immersion of gaseous $CF_x$ (mainly $C_2F_4$) (Supplementary Fig. 6) from PTFE into melted Li, providing a large reaction interface, is favorable for self-propagation (Supplementary Figs. 7 and 8). Consequently, MGC was obtained after the removal of excess Li metal and LiF with

methanol and diluted hydrogen chloride. It is noteworthy that the entire reaction can be finished in less than one millisecond (Supplementary Fig. 9) and requires no external energy input throughout the synthesis process, which opens up an economical way to produce porous graphitized carbons, verifying its priority over traditional methods as shown in Supplementary Fig. 1. Another important characteristic of the lithiothermal strategy is its feasibility for industrial-level material preparation, which potentially to be realized through a tunnel kiln (Supplementary Fig. 10). To demonstrate the process of the microscopic characterization for MGC and show the detailed combustion process close to the experimental observations, the temperature distribution was simulated numerically using COMSOL multiphysics simulation (details in the Supplementary Methods, Supplementary Fig. 11). The simulation results displayed an obvious high-temperature reaction zone (>3000 K). The high temperature can also be confirmed by the melting of niobium metal with a melting point higher than 2700 K (Supplementary Fig. 12).

The MGC shows thin corrugated sheets with abundant pores (Fig. 1B). High-resolution transmission electric microscopy (HR-TEM, Fig. 1C) clearly reflects well-defined lamellas with onion-like features and average $d$-spacing of 0.36 nm, validating the graphitic structure of MGC. The formation of carbon onion structure is attributed to the ultrahigh temperature, which will enable the graphitic sheets to curve and merge to minimize the number of dangling bonds[21]. The reaction is also conducted in air to obtain oxygen-rich MGC (OMGC). Since the PTFE powder is tightly wrapped by the Li foil, oxygen is too limited to burn the reactant. The oxygen content of OMGC is 15.5 at.% (Supplementary Tables 1 and 2). The OMGC shows similar onion-like features with MGC, but shows local turbostratic graphitic microstructures (Supplementary Fig. 13). The porous structures of the two carbons were investigated by the nitrogen adsorption/desorption

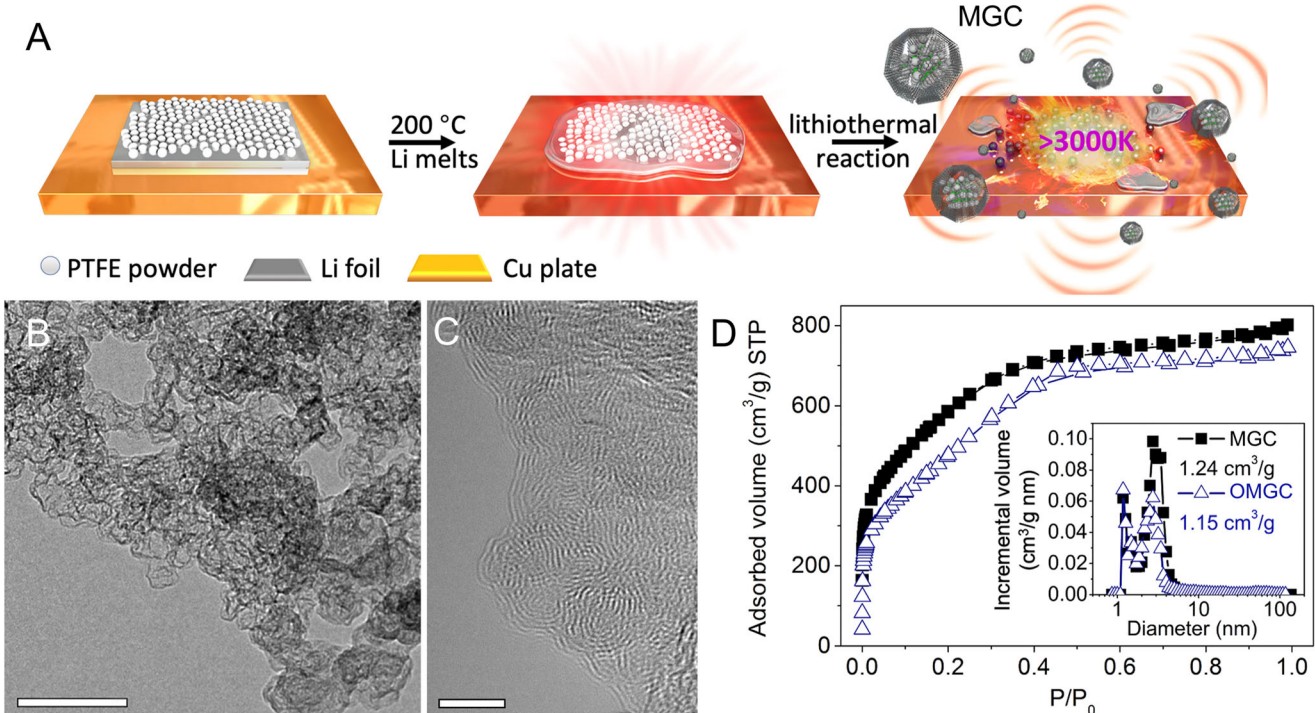

**Fig. 1 | Structure of highly porous graphitized carbon. A** Schematic the lithiothermal synthesis of highly porous graphitized carbon at ultrahigh temperature generated by the reaction between PTFE powder and molten Li. **B** TEM and **C** HR-TEM of highly porous graphitized carbon, while the former demonstrates highly porous microstructure and connected pore channels for the products, there is variation in focus across the image because of the sloped nature of the sample and

changes in sample thickness, the later shows that the products consist of a cluster of nano onions with the lattice spacing between the bent graphitic layers close to 0.36 nm. The image exhibits the presence of a dense network of nanometer-scale pores surrounded by highly graphitized multi-carbon layers. Scale bar: 100 nm (**B**) and 5 nm (**C**). **D** N₂ adsorption/desorption isotherms and pore size distribution (inset) of the sample.

isotherms (Fig. 1D), which shows type I adsorption, implying rich of micropores[22]. The pore-size distribution analyzed by the non-local density functional theory (DFT) model demonstrates a dominant micropore size of ~1.2 nm and mesopore size of ~3.0 nm (inset of Fig. 1D and Supplementary Figs. 14 and 15). The diameter of LiF nanoparticles generated after lithiothermal reaction is 2.8–4.9 nm, matching well with voids among carbon onions, which demonstrates these small mesopores are just the void occupied by LiF nanocrystals. The spherical aberration-corrected HR-TEM image (Supplementary Fig. 15) exhibits tremendous open channels in MGC with diameters ranging from 0.85 nm to 1.25 nm. This cross-coupled structure is totally different from the dead-end pores typical in activated carbon and other highly porous carbons[23]. These inter-connected channels not only contribute high surface area but can also serve as pathways for fast ion transport. The unique pore structure is contributed by the ultrashort condensation process for carbon atoms (Supplementary Fig. 16) and the migration of LiF with low boiling point (1681 °C) under ultrahigh temperature, inferred from clearly distinguished LiF in transmission electric microscopy (TEM) image (Supplementary Figs. 5 and 17), which is distinct from LiF buried in porous carbon through gentle defluorination of PTFE with $Li_2CO_3$ (Supplementary Fig. 18). In despite of the porous carbon from defluorination of PTFE reported in the 1990s using alkali metals or their amalgams under moderate conditions in Ar[24,25], moderate defluorination of PTFE produces chemically active polyynes and the fluoride, the porous carbon was subsequently obtained by crosslinking polyynes under extra acid or heat treatment. Unfortunately, large amounts of oxygen (up to 20%) and hydrogen are bound chemically to the active carbon skeleton in an acid treatment process[24], producing an inferior graphitized structure. When heat treatment is applied instead of acid leaching out, some macropores form due to the melting of the fluoride, leading to a wide pore size distribution for the porous carbon obtained[24]. In recent years, an artificial solid electrolyte interface was also constructed for the protection of alkali metal anodes by in situ reaction of molten alkali metal with PTFE, mainly focusing on the role of its high $Li^+$ conductivity for the products (LiF or KF/carbon composites)[20,26].

X-ray diffraction (XRD) patterns agree well with the TEM image (Fig. 2A), in which the MGC shows a sharp peak at $2\theta = 25.6°$, slightly larger than the (002) peak ($2\theta = 22.5°$) of the commercial activated carbon (YP-50, provided by Shenzhen Kejing Star Technology Co., Ltd), which indicates d-spacing of 3.36 Å and suggests significantly elevated graphitization degree of carbon. Raman spectrum exhibits a pronounced 2D peak (2690 $cm^{-1}$) consisting of a single Lorentzian component (Fig. 2B and Supplementary Fig. 19) and shows obviously stronger G band than D band (G band, arising from planar $sp^2$-configured carbon atoms, D band, attributed to $sp^2$ carbon atoms with defects), further confirming its higher graphitization and formation of more ordered carbon structure[27]. After introducing oxygen-containing groups, there is a significant broadening of (002) peak width for OMGC (Fig. 2A), indicating that the oxygen atoms may prevent the graphitization via the formation of functional groups and cross-linking structure. The result was further verified by the decreased $I_G/I_D$ ratio from 1.10 to 0.86 in Raman spectroscopy (Fig. 2B). Electron energy loss spectroscopy (EELS) profile (Fig. 2C) is another complementary tool to quantify the amount of $sp^2$-hybridization. The peak at 285.5 eV is due to transitions from 1s state to $\pi^*$ states above the Fermi level[28]. This peak implies the existence of $sp^2$-bonded clusters. The broad peak from 290 ev to 310 eV is due to transitions from 1s to $\sigma^*$ states. The fractions of the $sp^2$ bonding for MGC and OMGC were estimated to be 76% and 63% according to the ratio of $\pi^*$ and $\pi^* + \sigma^*$, assuming that the bulk graphite reference spectrum is 100% $sp^2$-bonded. As a result, MGC may present excellent electrical conductivity of 459 S/m, which is five times higher than commercial activated carbon (10–100 S/m)[16]. OMGC also presents a high electrical conductivity of 235 S/m. In addition, X-ray photoelectron spectra (XPS) were collected for both MGC and OMGC

(Fig. 2D and Supplementary Figs. 20–22). The most prominent peaks at 284.6 eV and 533.2 eV in the survey spectra are designated as C 1s and O 1s, respectively, with the oxygen content of OMGC (15.5 at.%) higher than that of MGC (1.2 at.%). As shown in Supplementary Figs. 20A and 21A, five peaks denoting $sp^2$ C, $sp^3$ C, C–O, C=O, and O–C=O were deconvoluted with increasing binding energies. MGC and OMGC show 88 at.% and 73 at.% of $sp^2$ C content, respectively, which confirms the highly graphitized structure for MGC and OMGC.

Supercapacitors, which bridge the gaps between batteries and electrolytic capacitors, store energy by forming the electrical double layer on high-surface-area electrodes, generally porous carbons[29–31]. The power density of porous carbon electrodes is determined by the accessible surface areas under various charge/discharge rates[31], where both high electronic and ionic conductivities are indispensable for delivering superior power outputs[1–3]. The as-prepared MGC shows excellent high-rate capability in two-electrode symmetrical super-capacitors, with 1-ethyl-3-methyl-imidazolium tetrafluoroborate ($EMIMBF_4$) as the electrolyte. Figure 3 presents typical supercapacitor performance with a voltage range of 3.5 V. The cyclic voltammograms (CV) test (Fig. 3A) shows a quasi-rectangular shape from 0 V to 3.5 V over a wide range of voltage scan rates of 1–100 V/s, indicating nearly ideal and high-rate capacitive behavior. The scan rate of 100 V/s is more than two orders of magnitude higher than any result reported with alternative devices, including microdevices and microelectrodes[2,32–35]. The galvanostatic charge/discharge (GCD) curves at various current densities are shown in Fig. 3B. The specific capacitance is calculated from the discharge curves with values of 149 F/g, 121 F/g, and 110 F/g obtained at current densities of 0.1 A/g, 10.0 A/g, and 200.0 A/g, respectively (Fig. 3C). The voltage drop at the initiation of the discharge is 0.055 V at the current density of 100 A/g, exhibiting low equivalent series resistance (ESR) in the test cell. On the other side, an increase in scan rate leads to the enlargement of current density apparently, meanwhile, a quasi-rectangular shape could be maintained from 1 V $s^{-1}$ to 50 V $s^{-1}$, while a higher scan rate results in a significant distortion (Supplementary Fig. 23A). A series of GCD curves at current densities ranging from 0.1 A $g^{-1}$ to 200 A $g^{-1}$ for OMGC are displayed in Supplementary Fig. 23B. OMGC shows obvious voltage drop especially at large discharge current density, leading to its slightly inferior rate capability than MGC. The OMGC delivers a slightly higher specific capacitance of 177 F/g at 0.1 A/g, but it decreases rapidly to only 36 F/g with the increase of current density to 100 A/g.

The excellent supercapacitor performance is evidenced by electrochemical impedance spectroscopy (EIS). A frequency response analysis at 3.5 V over the frequency range from 100 k to 0.01 Hz yielded the Nyquist plots shown in Fig. 3D. All plots feature vertical curves, showing nearly ideal capacitive behaviors. Both MGC and OMGC cells show low ESR values (1.61 ohm and 2.06 ohm) and low charge transfer resistance ($R_{ct}$) of electrode/electrolyte interface (0.57 ohm and 0.26 ohm), lower than commercialized activated carbon[36,37]. The EIS shows a small time constant $\tau_0$ (the minimum time needed to discharge all the energy with an efficiency of >50%) of 31.5 ms for MGC, which is characterized as the inverse of the characteristic frequency $f_0$ ($\tau_0 = 1/f_0$) at −45° in the Bode phase plots (Fig. 3E). This can be attributed to the fact that the surface of inner pores of MGC is highly accessible to ion adsorption/desorption[38], which reduces response time to the external electrical field. Such small $\tau_0$ is comparable to the ultrahigh-power onion-like carbon electrodes, and much lower than that of the activated carbon- or other porous carbon-based electrodes. Moreover, those onion-like carbon electrodes typically use low mass loading of <1 mg/$cm^2$ with the areal capacitance of 0.4–3.67 mF/$cm^2$. In contrast, $t_0$ of 31.5 ms above is achieved at a high loading of 1.3 mg/$cm^2$ (156 mF/$cm^2$)[1,39–41]. After 100,000 charge/discharge cycles, the capacitance retention is 97.7%, which is comparable to the best results reported for carbon-based electrodes (Fig. 3F). OMGC presents a capacitance retention of 89.2%, also reaching the frontier level of

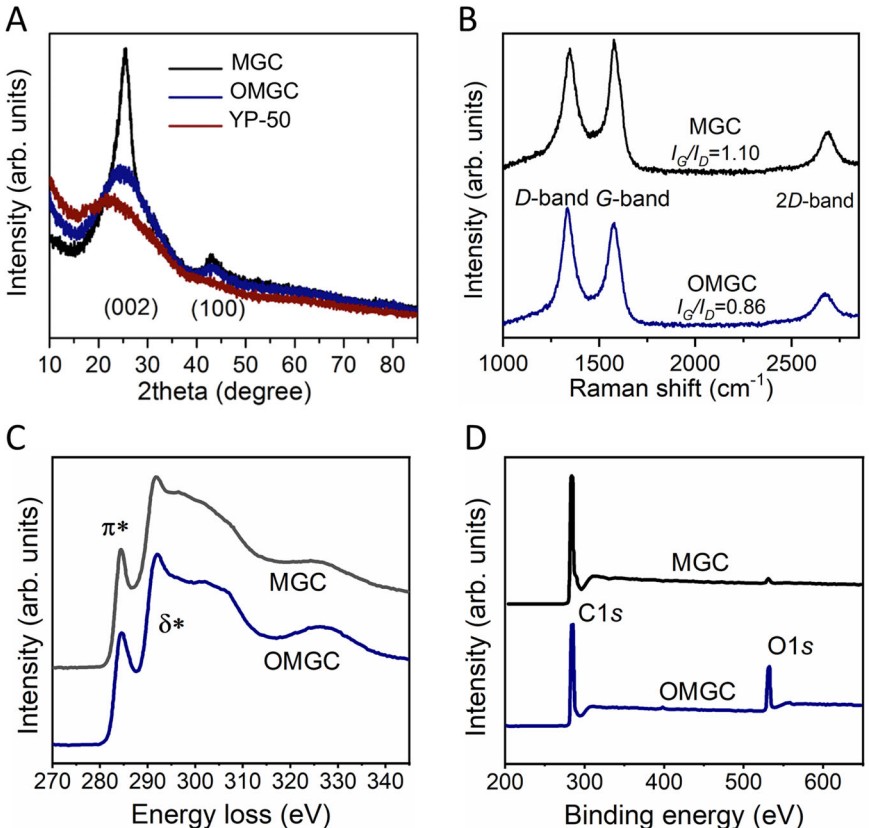

**Fig. 2 | Characterization of porous graphitized carbon. A** XRD pattern of porous graphitized carbon MGC, OMGC, and activated carbon. **B** Raman, **C** EELS, and **D** XPS spectra from MGC and OMGC.

carbon-based electrodes. It is also worth mentioning that the nitrogen adsorption/desorption isotherms for MGC before and after 100,000 cyclings almost overlap together (Supplementary Fig. 24). Besides, MGC-based electrode also presents favorable self-discharge performance (Supplementary Fig. 25). All these results validate the superb cyclic stability for MGC.

For comparison, PTFE powder and Li metal also reacted in a mass ratio of 3.75 with PTFE 5 wt% excess (MGC-5%). It is confirmed that there is fluorine left over in MGC-5% (Supplementary Fig. 26A–C). As a result, MGC-5% presents an electrical conductivity of 326 S/m, just inferior to 459 S/m for MGC, indicating that the fluorine atoms may prevent the graphitization via the formation of functional groups and cross-linking structure (Supplementary Fig. 26D, E). Besides, MGC-5% the micro–meso–macroporous texture with a dominant micropore distribution (1–2 nm or 2–4 nm) and a slight meso–macro pore distribution (10–100 nm) (Supplementary Fig. 26F). When evaluated in a two-electrode configuration in $EMIMBF_4$, MGC-5% presents specific capacitance of 140.6 F/g at 0.1 A $g^{-1}$, and 42.3 F/g at 200 A/g, and holds satisfactory capacitance retention of 92.1% at 10 A/g after 100,000 cycles, slightly inferior to MGC (Supplementary Fig. 27). Therefore, the porous graphitized carbon MGC from the complete defluorination reaction between Li metal and PTFE shows the optimum performance, that is probably because the complete defluorination of PTFE is favorable for the self-propagation reaction.

The merit of this material relative to the state-of-the-art battery and supercapacitor materials was evaluated in Supplementary Fig. 28 and Table S3. It is shown that the nanocarbon electrodes, such as nano-onion, can ensure the cells obtain capacitance at scan rates of as high as 5–200 V/s and get $f_0$ of as high as 30–100 Hz[16]. However, all these superior rate capabilities are based on electrodes with low areal mass loading (ultrathin thickness) of active materials, i.e., typically less than 1 mg/cm², limiting the practical applications. Here 1.3 mg/cm² loading

was used, which is a typical value for commercialized-level activated carbon electrodes[42]. To pack more energy and power into the device, we increased the mass loading to the limit of minimal sacrificing electrochemical performance[43]. Even up to 8.5 mg/cm², the gravimetric specific capacitance of the symmetric electrochemical cell changed minimally (Supplementary Fig. 29).

The excellent supercapacitor performance is further demonstrated in practical pouch cells with ~50 F. The pouch cell uses two pairs of MGC electrodes with internal parallel architecture and mass loading of 2.8 mg/cm² (Fig. 3G and Supplementary Fig. 30), which shows a high capacitance of 57 F when charged to 3.5 V. The specific capacitance remains at 142 F/g, only 5% less than the results at coin cell scale (Fig. 3C), indicating the potential of MGC to be scaled up for commercial applications. Notably, the MGC capacitor delivers 40.4 F at 20.0 A (corresponding to 100.2 F/g at 50 A/g), much higher than that (14.9 F at only 2.5 A) of YP-50, demonstrating its superiority in both capacitance and rate capability (Fig. 3G and Supplementary Fig. 30). This high capacity supercapacitor with good performance permits exciting opportunity for practical power applications with both improved energy and power performances.

Besides, the excellent capacitive performance of MGC is also attractive for CDI, which is a promising desalination technology to obtain freshwater[7]. By applying a potential between the two electrodes, the salt ions are stored in the electrical double layer of the electrode, thus clean drinking water is obtained. Porous carbon with both high SSA and electronic conductivity is critical for superior CDI performance. The MGC exhibited higher desalination capacity and faster rate compared with one best commercial capacitive activated carbon in 500 ppm NaCl aqueous solution (Supplementary Fig. 31).

Particularly, different from the porous carbon with lots of closed pores prepared from the "top–down" synthetic route[23], MGC was synthesized via the "bottom–up" route, the instantaneously (<1 ms)

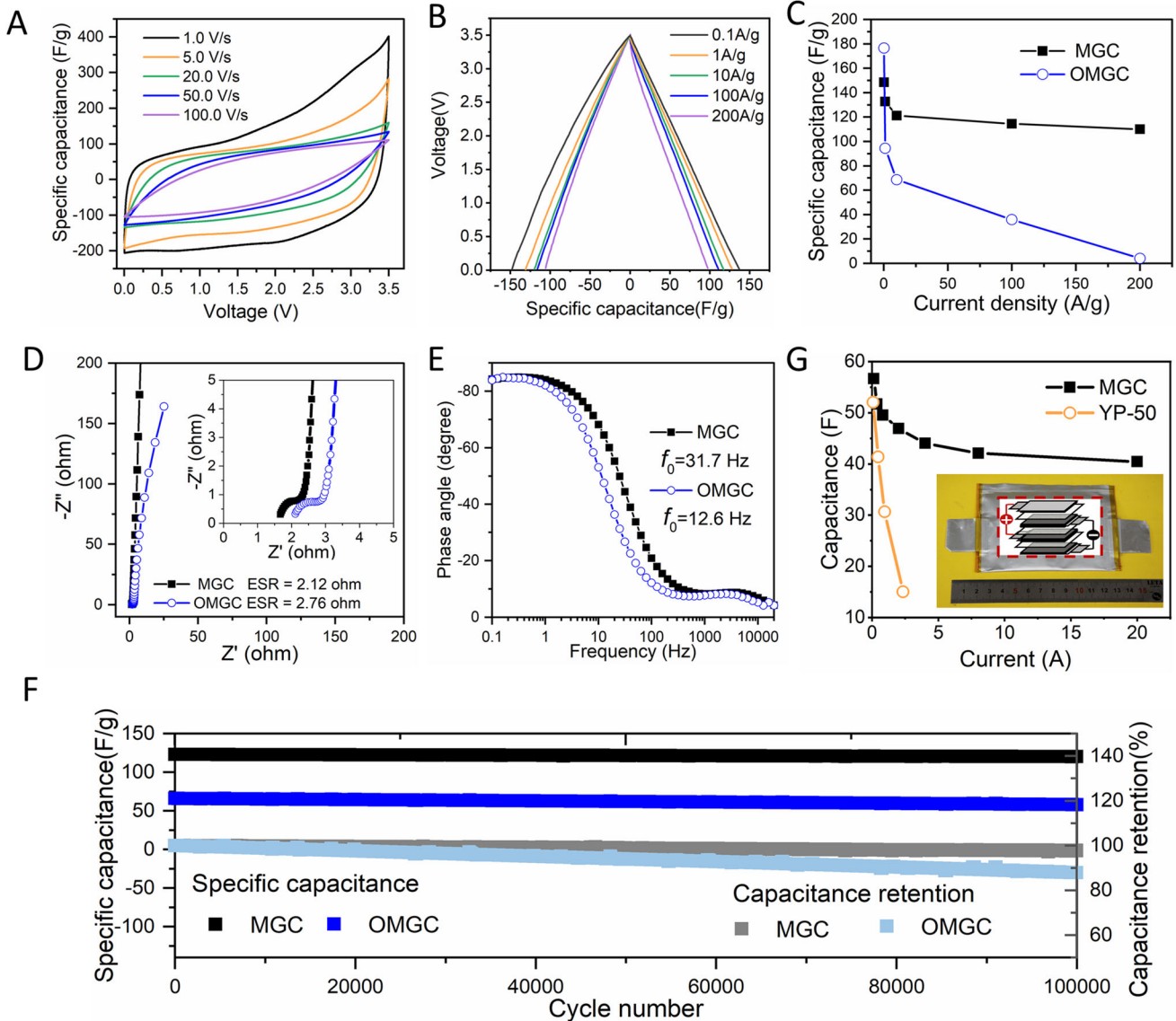

**Fig. 3 | Supercapacitor performance of MGC in EMIMBF₄ electrolyte. A** CV curves from 1 V/s to 100 V/s of MGC, which can sustain high scan rates, like electrolytic capacitors. **B** GCD curves at 0.1–200 A/g for MGC. **C** Variation of specific capacitance with charge/discharge current density of MGC and OMGC. **D** Nyquist plots for MGC and OMGC capacitors were taken at 3.5 V. The insets show magnification of the high-frequency region. **E** Phase angle versus frequency for MGC and OMGC. **F** Capacitance retention for MGC and OMGC capacitors cycled at 10 A/g to 3.5 V. **G** Variation of specific capacitance with charge/discharge current density of soft-package supercapacitor based on MGC and YP-50 in EMIMBF₄. Their corresponding capacitance is 57 F and 52 F. Inset is the digital images of a soft-package device with an internal parallel structure.

gas-phase condensation tends to form cross-coupled open channel under ultrahigh temperature, which serves as ion-highways for fast ionic transport without block and dead ends. Benefiting from the ultrahigh temperature, the excellent electrical conductivity for MGC can also be ensured by its highly graphitized degree. In brief, distinctive interconnected open channels with low oxygen content as well as highly graphitic degree guarantee fast ion and electron transportation for MGC.

Further mechanistic studies unveil that excellent performance originates from two factors, the high electrical conductivity of carbon and the open porous structure. First, the high electrical conductivity of carbon decreases the internal resistance of electrodes (low ESR values of 1.61-ohms for MGC). Second, the open porous structure provides well-defined pathways for rapid ion transport. The ultrahigh charge transfer rate features for MGC are validated by the COMSOL multiphysics simulation of BF₄⁻ flux distribution. (mimicking ion transportation behavior in porous carbon) (details in the Supplementary

Methods). As shown in Fig. 4A, regions in red are clearly visible, demonstrating a highly enhanced ion flux in interconnected pores over that of hierarchical pores (Fig. 4B). Consequently, the enhanced ion flux for interconnected pore contributes to better rate performance than the hierarchical porous carbon (Fig. 4C).

Besides, the interaction between electrolyte ions and porous carbon can influence ion transport as well. Herein, the theoretical calculation of adsorption energy between BF₄⁻ and carbon basal layer with various defects, including holes and one or two carbonyl groups is well conducted (Fig. 4D). Compared with the intact basal layer, the hole defects seem to have little effect on adsorption energy. In contrast, the adsorption energy between one and two carbonyl groups and BF₄⁻ increased sharply from −0.46 eV to −1.12 eV, and −1.30 eV, which indicates that the interaction of oxygen-containing functional groups with BF₄⁻ is strong, and may retard the OMGC response to fast charge/discharge behavior. Therefore, the absence of an oxygen functional group is preferred to enhance electrolyte ion transport for porous

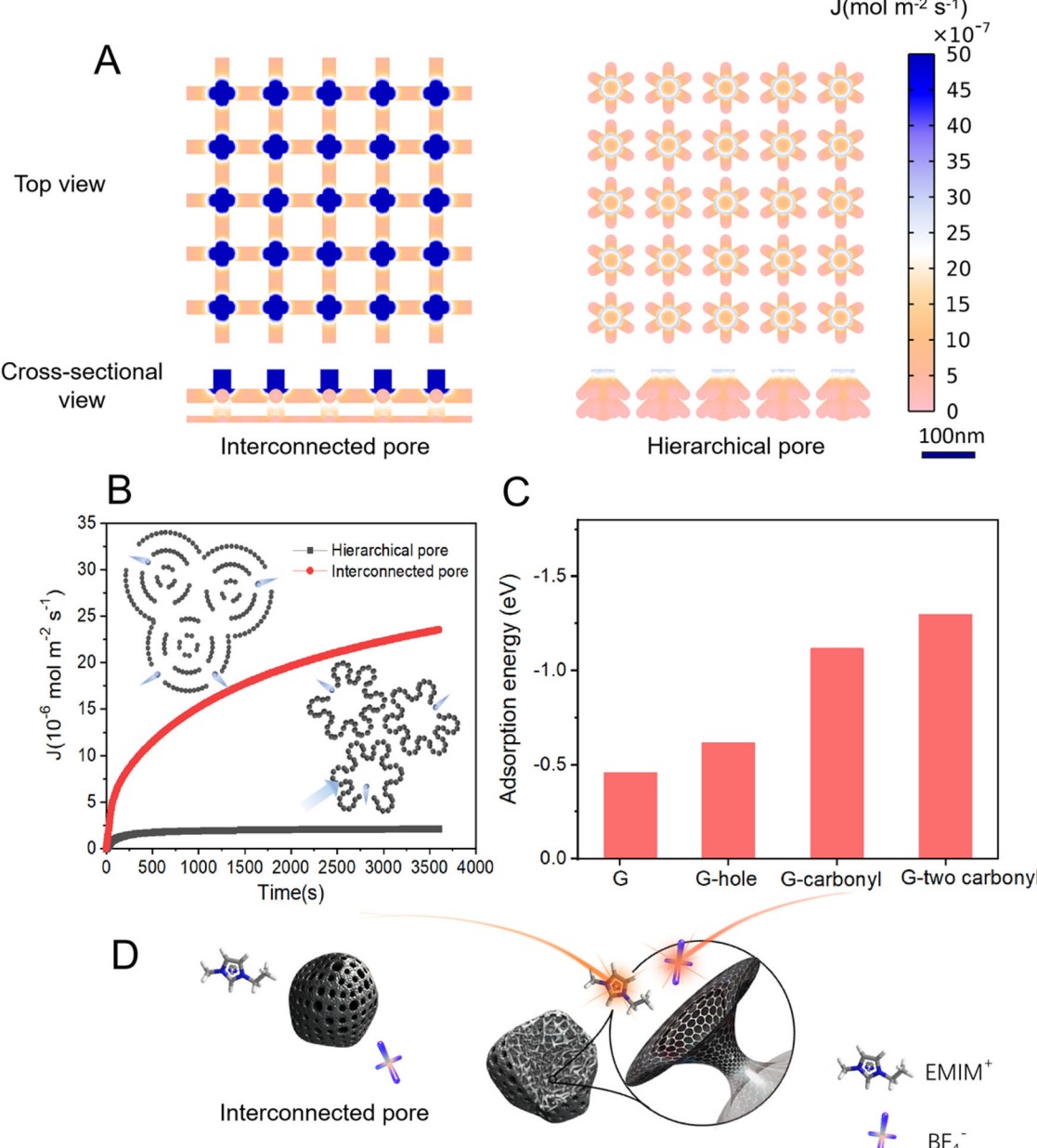

**Fig. 4 | Origin of ultrahigh rates. A** Simulation results of the spatial ion flux distribution of electrolyte ions in the two different pores of interconnected pore and hierarchical pore from the top view and cross-sectional view. **B** The total electrolyte ion flux for interconnected pores and hierarchical pores. **C** Calculated adsorption energy of $BF_4^-$ ion with various defects, including hole, carbonyl, and two carbonyls, on carbon basal layer. **D** Schematic illustration of the ion transport in nanopores of MGC. Because of the low adsorption energy between ions and functional group-free, highly graphitized pore walls of MGC, ions can diffuse along the inner surface of MGC pores with much faster charge transfer dynamics, like transport on a lubricated surface and ensure an ultrahigh charge/discharge rate capability.

carbon. MGC possessed a low oxygen content, thus, $BF_4^-$ ions tend to move freely on the "nonstick surface" of MGC's inner pores.

## Discussion

In summary, we demonstrate a scalable and ultrafast lithiothermal process to prepare a highly porous, and low oxygen content graphitized carbon. The process obviates the time-consuming and limited yields in conventional top–down and bottom–up production ways and derives their advantages of high reactive carbon atoms with high spatial density and high product quality. The cross-coupled open pore channels of MGC supply as diffusion pathways to facilitate the ion transport and migration in the electrode bulk. Typically, serving as the electrode material for supercapacitor in an electrolyte of $EMIMBF_4$, the

MGC electrode achieves a desirable rate capability, retaining more than 90% of the capacitance when the current density increases from 10.0 A/g and 200.0 A/g. Impressively, MGC also delivers a superb capacitance retention of 97.7% after 100,000 cycles, suggesting its great potential as a high power supply. Furthermore, MGC also demonstrates its superiority over commercial activated carbon in pouch cells with an internal parallel architecture, showing a specific energy of 47 $Wh/kg_{MGC}$ and a specific power of 175 $kW/kg_{MGC}$. Characterized by ultrahigh temperature and solid-state reagent, this methodology is extended to create highly graphitized porous carbon by other metal elements (Na, K, Mg, Al, etc.) (Supplementary Figs. 32–33) and other nonmetal main group element nanostructures (B, Si, etc.) (Supplementary Figs. 34–36). Our work can also help to

illuminate astrophysical processes near carbon stars or supernovae that result in nano-carbon cluster formation throughout the universe. This high performance, together with a convenient preparation process, makes the lithiothermal method a promising strategy for practical application towards electrochemical energy storage and delivery.

## Methods

### Synthesis of MGC

The MGC was prepared by a simple lithiothermal reaction between PTFE powder and molten Li metal. PTFE powder and Li metal reacted in a mass ratio of less than 3.57 (stoichiometric mass ratio). To show the optimum condition, PTFE powder, and Li metal also reacted in a mass ratio of 3.75 with PTFE 5 wt% excess (MGC-5%). Specifically, in a sealed chamber filled with Ar (here we use glove box), PTFE powder (analytical grade) was enveloped into a Li bag (3 cm × 2 cm), and clamped using two Cu plates with 1 MPa in a plate-type thermocompressor. Then Cu plates were heated to 200 °C followed by repeated friction, the reaction was immediately initiated and completed within a millisecond. After the reaction, the collected powder was rinsed in methanol and 1 M HCl to completely remove the remaining Li metal and LiF, and MGC was finally obtained.

For comparison, we also prepared OMGC in the same way in the air. The reaction was conducted in a heat press with two independent temperature control platforms, bearing a maximum temperature of 500 °C.

### Characterization

The scanning electron microscopy observation and energy dispersive X-ray spectroscopy mapping were conducted on FEI XL30 Sirion operating at 5 kV. TEM and HR-TEM characterization were taken on FEI Tecnai G2 F30. The accelerating voltage was set as 200 kV. Spherical aberration-corrected HR-TEM was performed on a JEOL JEM-2100F. EELS was carried out on Spherical aberration-corrected JEOL JEM-2100F TEM with commercial graphite powder (325 mesh) as reference. For TEM analysis, samples were prepared by dipping a drop of MGC/ethanol dispersion onto a holey carbon-coated copper grid and dried at room temperature for 1 h to completely evaporate the solvent.

XRD patterns were recorded on a multifunctional Bruker D8 advance X-ray diffractometer with monochromatic Cu K$\alpha$ radiation ($\lambda = 1.54060$ Å) at a scanning rate of 10 degrees per minute. Raman measurements were performed on LabRam HR-800, Horiba Jobin Yvon with an excitation wavelength of 532 nm. XPS was collected on an electron spectrometer (ThermoFisher ESCALAB 250 Xi) to analyze surface elemental composition. Binding energies were calibrated using containment carbon (C 1$s$ = 284.8 eV).

The nitrogen adsorption/desorption measurements were carried out at 77 K on Micromeritics ASAP 2020. The surface area of the samples was determined from the Brunauer−Emmett−Teller (BET) method. The pore volumes and pore size distributions were derived from the DFT model.

Differential scanning calorimetry was performed on a TA Auto Q20 Instrument in nitrogen (New Castle, DE, USA). Samples (ca. 5 mg) were sealed in a high-pressure hermetic crucible with an empty sealed hermetic pan serving as the reference. Thermal transitions were reported on the third heating cycle. Samples were heated/cooled at a rate of 5 °C/min.

### Electrochemical characterization

The cell supercapacitors were composed of two symmetrical working electrodes sandwiched by a separator and the electrolyte of EMIMBF$_4$. For the preparation of working electrodes, 85 wt% of active material powder (MGC, OMGC, or YP-50), 10 wt% of carbon black, and 5 wt% of PVDF were mixed with a few drops of N-methyl-2-pyrrolidone to form a slurry, which was then uniformly coated onto aluminum foil current collectors by doctor blading. The prepared electrodes were dried at

120 °C for 12 h and punched into round disks with a diameter of 16 mm. Two identical disk electrodes and a cellulose separator (NKK TF4050) were sandwiched into a 2032 coin-typed cell for further testing.

The soft-package supercapacitors were fabricated based on the electrodes prepared as described above. The electrodes were compressed under 1 MPa pressure with an average thickness of 110 μm for active materials, then cut into 8 cm × 6 cm rectangles. The loading mass of active materials on each side of the electrodes was all the same with an average mass loading of 2.8 mg cm$^{-2}$ for MGC and 2.9 mg cm$^{-2}$ for YP-50. The electrodes were assembled in an internal parallel configuration. First, the two end electrodes were prepared by one-side loading electrode materials onto aluminum foils as described above, and then the bipolar electrode in the middle was prepared similarly by double-side loading electrode materials onto aluminum foils. Two separators were settled between the electrodes. Finally, the device was infused with EMIMBF$_4$ electrolyte (8 mL) and sealed with aluminum plastic film.

CV tests and EIS tests were performed on the Solartron 1280 test station under ambient conditions. The GCD tests were conducted on the Arbin BT2000 battery testing system. The potential range for CV and GCD experiments was 0–3.5 V. The EIS test was carried out in the frequency range of $10^5$–$10^{-1}$ Hz with 5 mV amplitude corresponding to open circuit potential.

The gravimetric ($C_{2,g}$, F g$^{-1}$) and areal ($C_{2,a}$, μF cm$^{-2}$) specific capacitances of the electrode materials in the two-electrode system were obtained from the GCD curves. The specific capacitance of the electrode can be calculated according to the following equations:

$$C_{2,g} = \frac{2I \times \Delta t}{m \times \Delta V} \quad (1)$$

$$C_{2,a} = \frac{100 \times C_{2,g}}{S_{BET}} \quad (2)$$

where $I$(A) is the current, $\Delta t$ (s) is the discharge time, $\Delta V$ (V) is the potential drop during the discharge process, $m$(g) is the mass of active material in a single electrode, and $S_{BET}$(m$^2$ g$^{-1}$) is the SSA of the carbon sample obtained from the BET method.

Gravimetric energy ($E_g$, Wh kg$^{-1}$) and power ($P_g$, W kg$^{-1}$) density of the symmetric supercapacitor cell based on the mass of active material were calculated according to the following equations:

$$E_g = \frac{1}{2} \times \frac{C_{2,g}}{4} \times \Delta V^2 \times \frac{1}{3.6} \quad (3)$$

$$P_g = \frac{E_g}{\Delta t} \times 3600 \quad (4)$$

where $C_{2,g}$ (F g$^{-1}$) is the gravimetric specific capacitance of a single electrode based on the active material, $\Delta V$ (V) is the potential change in discharge, and $\Delta t$ (s) is the discharge time.

### Desalination test

The capacitive desalination performance of the CDI system was evaluated with a 2.0 × 0.5 × 2.0 cm$^3$ cell including two 2.0 × 2.0 cm$^2$ symmetric electrodes plated at each side, with a 0.5 cm distance between them. The electrolyte was 500 ppm NaCl aqueous solution. MGC and YP-50 were used as CDI electrodes with the areal mass loading of 9.1 mg/cm$^2$ and 10.5 mg/cm$^2$ (both 300 μm thick), respectively. The MGC and AC electrodes were fabricated according to a similar procedure to those for electrochemical evaluations. In the charging step (ion-capturing step), the ions in the solution were captured by the potential (0.8 V, 1.0 V, and 1.2 V) applied between the two electrodes. The concentration of NaCl solution was measured by an ionic conductivity meter (UniCond 2-E; METTLER TOLEDO, USA). The salt

adsorption capacity (SAC, $mg\,g^{-1}$) was calculated from the following equation:

$$SAC = (C_0 - C)V/m \tag{5}$$

where $C_0$ ($mg\,g^{-1}$) is the initial NaCl concentration, $C$ ($mg\,g^{-1}$) is the concentration at a certain time during the process of adsorption and desorption, $V$ is the total volume of the NaCl solution (2 mL), and $m$ represents the total mass of the active components including anode and cathode in CDI device[44,45].

## Data availability

All relevant data that support the findings of this study are presented in the manuscript and supplementary information file. Source data are provided with this paper.

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

## Acknowledgements

We thank Q. Fu from Peking University for help with the COMSOL multiphysics simulation. This work was supported by the Natural Science Foundation of China (22075320 to H.Z. and 52202336 to H.M.Z.).

## Author contributions

H.Z. initiated the project. H.Z. and F.L. conceived the experiments. H.M.Z. conducted the experiments and wrote the paper. H.Z., J.Q., J.P., G.C., X.H., S.M., X.Z., H.M. and X.F. assisted with the data analysis; and B.Z., L.W. and Z.L. performed the DFT simulations, and all the authors were involved in revising the manuscript.

## Competing interests

The authors declare no competing interests.
