## [Peer Review File · Nature Communications]

Sub-millisecond lithiothermal synthesis of graphitic meso-microporous carbonREVIEWER COMMENTS

Reviewer #1 (Remarks to the Author):

This manuscript describes a novel lithiothermal method to synthesize porous carbon with impressive electrochemical reactivity and excellent electrical conductivity. Notably, the meso-microporous graphitized carbon (MGC) electrode achieves a remarkable rate capability, retaining more than 90% of the capacitance when the current density increased from 10.0 and 200.0 A/g. Together with its feasibility for scaling up materials synthesis, this method solves the major issues that porous carbon is facing. The MGC material enables one to realize high-performance and cost-effective supercapacitor devices.

I suggest the work be accepted for publication after addressing the the following concerns.

1. In Figs. 2 B-D, arbitrary unit (a.u.) is used in y-axis. Does this mean the absolute value of coordinates is meaningless? In Fig. 3 D, the title of y-axis should be corrected. In Fig. 3E, τ_0 should be replaced with f_0 .
2. To ensure the consistency and repeatability of the MGC synthesis, the "Synthesis of MGC" section should be detailed. For example, the mass ratio between PTFE powder and lithium metal should be provided.
3. In recent years, flash Joule heating process has been applied to synthesize high performance nanomaterials. This technology is a highly efficient and fast heating method that can heat up to 3000-4000 °C within 1 second. The difference between the flash Joule heating technology and the lithiothermal method needs be discussed.
4. The description of the state of the art in the introduction is brief.
5. DSC in Fig.S8 is used to confirm the ultrafast exothermic reaction between lithium metal and PTFE. However, the detailed description of the characterization is missing in the "Characterization" section. The DSC characterization method in this study may be different from traditional ones because of its ultrafast and ultrahigh exothermic behavior for the lithiothermal method.
6. The reaction was also conducted in air to obtain oxygen rich MGC (OMGC). While the PTFE powder was wrapped by the Li foil, a 200 °C thermal stimulus was applied. The author needs to explain how he attempted to avoid the oxidation of lithium metal in air. It is suggested to provide the details of the reaction setup.
7. The material preparation process as illustrated in Figs. S29 and S31 is missing.
8. The author is suggested to conduct a self-discharge measurement as the self-discharge process may be of a limiting parameter for supercapacitors.

Reviewer #2 (Remarks to the Author):

The authors designed and synthesized a highly graphitized meso-microporous carbon (MGC) by intensive and ultrafast exothermic reaction between polytetrafluoroethylene (PTFE) powder and molten Li metal, which shows high surface area and good graphitized crystallinity. When MGC is applied to supercapacitors, it can deliver good electrochemical performance and long stability. Although the concept is interesting, I have found several major issues that need to be addressed.

1. Although the authors used COMSOL simulation and the melting of niobium metal to investigate the reaction temperature of lithiothermal reaction, I still suggest that MGCs prepared with different reaction conditions should be obtained and characterized to show the optimum conditions.
2. The authors prepared and compared MGC and OMGC. However, additional characterization about MGC and OMGC should be provided, for example, the electrical conductivity of OMGC and the BET specific area of OMGC, and so on.
3. The authors should explain in-detail all characterizations and figures, or present it in supporting information, for example, why did the authors give XPS results of different samples? In Ar, there are a lot of fluorine left over, while the XPS spectra of MGC and OMGC did not show F1s spectra. Why?
4. For supercapacitors, the authors used with 1-ethyl-3-methyl-imidazolium tetrafluoroborate (EMIMBF₄) as the electrolyte. Why? I wonder if the MGC could show good electrochemical performance in an acidic or alkaline electrolyte.
5. Check that all parts, insets, and Figures caption are mentioned, for example, Fig. S5. Check that the labels of the axes in all graphs are correct, for example, Fig. 3D.
6. For comparison, the authors should give the CV curves, charge/discharge curves and long-cycle performance of OMGC.

Reviewer #3 (Remarks to the Author):

The study presents development of Highly graphitized meso-microporous carbon materials by instantaneously gas-phase condensation that tends to form cross-coupled open channel under ultrahigh temperature, as electrode for supercapacitor application.

the study is well supported by detail physiochemical and electrochemical characterizations.

the study and results are significant to the field, and sufficient details have been provided.

the manuscript can be considered for publication with minor revision.

1. page 5, "Raman spectrum exhibits pronounced 2D peak (2690 cm⁻¹) of turbostratic graphite,.....". please note 2D peak is a measure of the stacking of layers in graphitic carbon. the statement must be corrected.
2. Fig. 3 (B). please correct the caption of the X axis in GCD curve. similar correction is also required in SI Fig. S24 (B)

3. Page 14, Synthesis of MGC..."and clamp it using two Cu plates with 1 MPa.". please clarify the statement.

4. There are a few grammatical errors, hence the manuscript must be thoroughly checked.

Dear Reviewers:

Thank you for the comments concerning our manuscript entitled “Sub-millisecond lithiothermal synthesis of graphitic meso-microporous carbon”. Those comments are most valuable and helpful for revising and improving our paper, as well as providing the important guidance to our research. We have modified the manuscript accordingly, and listed the detailed corrections below point by point for each reviewer. Moreover, all revised portion has been marked in yellow in the revised manuscript.

The main corrections and the responses to the reviewers’ comments are as follows.

Reviewer #1 (Remarks to the Author):

This manuscript describes a novel lithiothermal method to synthesize porous carbon with impressive electrochemical reactivity and excellent electrical conductivity. Notably, the meso-microporous graphitized carbon (MGC) electrode achieves a remarkable rate capability, retaining more than 90% of the capacitance when the current density increased from 10.0 and 200.0 A/g. Together with its feasibility for scaling up materials synthesis, this method solves the major issues that porous carbon is facing. The MGC material enables one to realize high-performance and cost-effective supercapacitor devices. I suggest the work be accepted for publication after addressing the following concerns.

1. In Figs. 2 B-D, arbitrary unit (a.u.) is used in y-axis. Does this mean the absolute value of coordinates is meaningless? In Fig. 3 D, the title of y-axis should be corrected. In Fig. 3E, τ_0 should be replaced with f_0 .

Response: Thank you for your careful review. We have modified Fig. 2B-D, Fig. 3D and Fig. 3E. Besides, all figures in this manuscript have been checked in detail.

The related changes have been marked with a yellow background in Fig. 2B-D, Fig. 3D and Fig. 3E.

2. To ensure the consistency and repeatability of the MGC synthesis, the “Synthesis of MGC” section should be detailed. For example, the mass ratio between PTFE powder and lithium metal should be provided.

Response: Thank you for your careful suggestion. PTFE powder and Li metal reacted in a mass ratio of less than 3.57 (stoichiometric mass ratio). To show the optimum condition, PTFE powder and Li metal also reacted in a mass ratio of 3.75 with PTFE 5wt% excess.

The related changes have been marked with a yellow background in the section of **Synthesis of MGC in Methods**.

3. In recent years, flash Joule heating process has been applied to synthesize high performance nanomaterials. This technology is a highly efficient and fast heating method that can heat up to 3000-4000 °C within 1 second. The difference between the flash Joule heating technology and the lithiothermal method needs be discussed.

Response: Thank you for your inquiry regarding the difference between the flash Joule heating technology and the lithiothermal method.

The lithiothermal method in this manuscript is a self-propagating ultrahigh temperature synthesis process from the reaction itself without extra energy input. Ultra-high temperature makes the porous carbon highly graphitized, while pore forming agent LiF in-situ formed in the fleeting moment ensures uniform pore size distribution (1~3nm). The strategy seems made to order for preparing cross-coupled porous and highly graphitized carbon. Joule heating focuses mainly on ultra-fast heating to high-temperature, which can be widely applied to synthesize nano-materials. In addition, for Joule heating technology, a high power supply is required, and there is a strict requirement for the heating medium to achieve a high heating rate ($\sim 10^4$ °C/min), high calcination temperature and high cooling rate ($\sim 10^3$ - 10^4 °C/min). Therefore, Joule heating technology currently suffers from relatively low yield and higher cost. To obtain high quality products with good homogeneity, raw materials are specially treated to obtain high conductivity, and Joule heating set up need to be designed elaborately.

4. The description of the state of the art in the introduction is brief.

Response: Thank you for your constructive suggestion. We have extended the description of the state of the art in the introduction.

Laser scribing is a new strategy based on the photothermal and photochemical reactions between carbon precursors and laser spot, which opens a pathway for graphene-based electrode materials with high conductivity¹⁷. While the limited SSA of laser scribing graphene limits its application as electrodes for supercapacitors.

It is found that the halogen elements (F, Cl) connecting to vinylidene units are highly reactive so that dehalogenation can take place at room temperature in the presence of strong inorganic alkaline. Dehalogenation-based carbonization based on halogenated polymers, such as polyvinyl fluoride (PVDF), polyvinylidene chloride (PVDC), can be used to prepare high SSA porous carbons^{18, 19}. However, pore size distribution for the porous carbon obtained is relatively wider, and graphitization need to be improved.

The related changes have been marked with a yellow background in the section of **Introduction**.

5. DSC in Fig.S8 is used to confirm the ultrafast exothermic reaction between lithium metal and PTFE. However, the detailed description of the characterization is missing in the “Characterization” section. The DSC characterization method in this study may be different from traditional ones because of its ultrafast and ultrahigh exothermic behavior for the lithiothermal method.

Response: Thank you for your careful review. Differential scanning calorimetry (DSC) was performed on a TA Auto Q20 Instrument in nitrogen (New Castle, DE, USA). Samples (ca. 5 mg) were sealed in a high-pressure hermetic crucible with an empty sealed hermetic pan serving as the reference. Thermal transitions were reported on the third heating cycle. Samples were heated at a rate of 5 °C/min.

The related changes have been marked with a yellow background in the section of **Characterization in Methods**.

6. The reaction was also conducted in air to obtain oxygen rich MGC (OMGC). While the PTFE powder was wrapped by the Li foil, a 200 °C thermal stimulus was applied. The author needs to explain how he attempted to avoid the oxidation of lithium metal

in air. It is suggested to provide the details of the reaction setup.

Response: Thank you for your inquiry about the details of the reaction. The reactant is placed between the heating plates of the hot press under pressure, and thus lithium metal is exposed into limited air. More importantly, the reaction proceeds quickly, which also reflects the advantages of this strategy.

Specifically, the reaction was conducted in a heat press with two independent temperature control platforms, bearing maximum temperature of 500 °C.

The related changes have been marked with a yellow background in the section of **Synthesis of MGC in Methods**.

7. The material preparation process as illustrated in Figs. S29 and S31 is missing.

Response: Thank you for your kind reminder. Si/C material was obtained as follows: Si and PTFE powder were compacted into reactant pillar under a pressure of 2 MPa, which was put into a crucible in the reactor of Fig. S33A. Then a 10 A current was exerted through the nickel chrome wire for 5 seconds to produce enough heat for the initiation of the reaction and Si/C material was obtained (Fig. S34).

Similarly, the reaction between Li metal and SnF₄ was conducted. SnF₄ was enveloped into Li bag, and clamp it using two Cu plates with 1 MPa. Then Cu plates were heated to 200 °C followed by repeated friction, the reaction was immediately initiated and completed within millisecond (Fig. S36).

The related changes have been marked with a yellow background in the caption of Fig. S34 and Fig. S36.

8. The author is suggested to conduct a self-discharge measurement as the self-discharge process may be a limiting parameter for supercapacitors.

Response: Thank you for your suggestion. Generally, speaking, the self-discharge is a spontaneous voltage drop process under the open circuit condition of supercapacitors, due to a high and negative Gibbs free energy (*ACS Nano* 2020, 14, 4916–4924). The undesired self-discharge led to the significant decay of the capacitance and a poor cycle-life. Here, the self-discharge performance of porous graphitized carbon was evaluated.

It is observed that the self-discharge rate of OMGC is larger than MGC. When the self-discharge voltage reduced from 3.5 V to 1 V, the required time for MGC (83.9 h) was prolonged in comparison to that of OMGC (50.3 h). On one hand, the most 1-10 nm pores favoring fast ion diffusion and formation of stabilized electric double layer structure would lead to a slower self-discharge rate, on the other hand, the oxygen groups on the surface of electrode materials will weaken the interaction between electrolyte ions and electrode surface and react with the electrolyte leading to the enhancement of the self-discharge. All these factors suppress the self-discharge behavior for MGC.

Fig. S25 Self-discharge curves of MGC and OMGC at the charge current density of 1A/g

Self-Discharge Measurements: Self-discharge tests were carried out using the LBT-21084. To certify the stability of the initial voltage, two-and-a-half charge/discharge cycles at the current density of 1A/g from 0 to 3.5 V were carried out prior to self-discharge.

The related changes have been marked with a yellow background in the text and Fig. S25.

Reviewer #2 (Remarks to the Author):

The authors designed and synthesized a highly graphitized meso-microporous carbon (MGC) by intensive and ultrafast exothermic reaction between polytetrafluoroethylene

(PTFE) powder and molten Li metal, which shows high surface area and good graphitized crystallinity. When MGC is applied to supercapacitors, it can deliver good electrochemical performance and long stability. Although the concept is interesting, I have found several major issues that need to be addressed.

Response: Thank you for your valuable feedback and suggestions. We appreciate your thorough review of our manuscript. We have carefully considered your comments and have taken the necessary steps to address the issues raised. We believe that these changes have significantly improved the quality and clarity of our manuscript. We are grateful for your constructive feedback, which has undoubtedly contributed to enhancing the overall impact of our work.

1. Although the authors used COMSOL simulation and the melting of niobium metal to investigate the reaction temperature of lithiothermal reaction, I still suggest that MGCs prepared with different reaction conditions should be obtained and characterized to show the optimum conditions.

Response: Thank you for your instructive suggestion. In our previous manuscript, PTFE powder and Li metal reacted in a mass ratio of less than 3.57 (stoichiometric mass ratio). In this condition, the defluorination of PTFE was complete. To show the optimum condition, PTFE powder and Li metal also reacted in a mass ratio of 3.75 with PTFE 5wt% excess (Fig. S26).

Fig. S26 Characterization of MGC-5%. XPS (A) C1s, (B) F1s and (B) O1s spectra of MGC-5%, (D) High-resolution TEM, (E) Raman spectra and (F) N₂ adsorption/desorption isotherms and pore size distribution (inset).

XPS measurement was applied to investigate surface element contents for MGC-5%. From the XPS spectra in Fig. S26, besides the intensive C 1s peak, oxygen and fluorine are also detected, confirming there is fluorine left over in MGC-5%. As shown in Fig. S26A, seven peaks denoting sp² C, sp³ C, C-O, C=O, O-C=O, covalently bonded C-F and semi-ionic C-F bond were deconvoluted with increasing binding energies. MGC-5% show 75 at.% of sp² C content, just slightly below 88% for MGC. MGC-5% presents electrical conductivity of 326 S/m, just inferior to 459 S/m for MGC. In F1s spectrum (Fig. S26B), the signals of semi-ionic fluorinated carbon chain (687.5 eV) and covalently bonded fluorine atoms (PTFE, 689.4 eV) can be characterized, respectively. The O 1s peak for MGC-5% shows similar configuration with MGC, which can be deconvoluted into three peaks corresponding to C=O groups, C-O groups, and O-C=O groups with increasing binding energies (Fig. S26C). High resolution-TEM (Fig. S26D) clearly reflects similar onion-like features with MGC and local turbostratic graphitic microstructures. In Raman spectra, the intensity of G band and D band is comparable with I_G/I_D ratio of 0.92, and obvious 2D peak was detected, indicative of its highly graphitization. But there is significant broadening of D and G band width for

MGC-5% (Fig. S26E), indicating that the fluorine atoms may prevent the graphitization via the formation of functional groups and cross-linking structure. The porous structures of MGC-5% were investigated by the nitrogen adsorption-desorption isotherms (Fig. S26F). A high nitrogen uptake in the low-pressure region ($P/P_0 < 0.01$), as well as a relative horizontal plateau in the medium-/high- pressure region demonstrates a typical microporous structure for MGC-5%. Besides, the slight but perceptible hysteresis loop between adsorption and desorption branch at $P/P_0 > 0.45$ can be classified as a sorption character for meso-macro porous materials. Additionally, pore size distributions based on DFT model are depicted in Fig. S26F, further confirming the micro-meso-macro porous texture with a dominant micropore distribution (1-2 nm or 2-4 nm) and a slight meso-macro pore distribution (10-100 nm). And MGC-5% shows the specific surface area of $1830 \text{ m}^2/\text{g}$, less than that of MGC. Therefore, the porous graphitized carbon MGC from the complete defluorination reaction between Li metal and PTFE shows the optimum performance, that is probably because the complete defluorination of PTFE is favorable for the self-propagation.

For comparison with MGC, the electrochemical properties of MGC-5% were estimated in a two-electrode configuration by cyclic voltammetry and galvanostatic charge-discharge techniques using EMIMBF₄ as electrolyte with the voltage range of 0-3.5 V (Fig. S27). When increasing scan rate from 1 V/s to 100 V/s, an enlargement of current density and a deviation from rectangular shape, meanwhile the quasi-rectangular shape could be preserved at a high scan rate of 50 V/s (Fig. S27A). Even at ultrahigh scan rate of 100 V/s, a minor distortion from rectangular shape was observed. Although there exists discernible capacity decaying from 20V/s to 50V/s compared to MGC, MGC-5% still shows good rate capability, because of the well-developed small mesopores for fast ion diffusion and the enhanced graphitization degree for good electron conductivity.

Charge-discharge curves of MGC-5% recorded at scan currents from 0.1-200A/g were displayed in Fig. S27B. As can be clearly seen, the shape of all the profiles is close to linear and symmetric triangular distribution, demonstrating a dominant electric double-layer performance. The specific capacitance reaches as high as 140.6 F/g at 0.1

A g^{-1} , and still retains 42.3 F/g at 200 A/g (Fig. S27C), holding a satisfactory capacitance retention of 30 %.

Additionally, long-term cyclability of MGC-5% was examined through galvanostatic charge-discharge technique at 10 A/g, and is illustrated in Fig. S27D. As shown, outstanding cycling performance has been achieved for MGC and MGC-5% samples. Particularly, MGC based cell exhibits an unprecedented cycling stability for maintaining 97.7 % capacitance retention. While, after 100,000 cycles, the capacitance retention for MGC-5% based supercapacitor is 92.1%. Though MGC-5% based electrode is slightly inferior than MGC in terms of electrochemical performance, this marvellous long-term electrochemical stability further confirms the excellent supercapacitive behavior of our carbon materials in practice.

Fig. S27 Supercapacitor performance of MGC-5% in EMIMBF₄ electrolyte. (A) CV curves from 1 to 100 V/s of MGC-5%, which can sustain very high scan rates. (B) Galvanostatic charge/discharge curves at 0.1 to 200 A/g for MGC-5%. (C) Variation of specific capacitance with charge/discharge current density of MGC and MGC-5%. (D) Capacitance retention for MGC-5% capacitors cycled at 10 A/g to 3.5V.

The related changes have been marked with a yellow background in the text and Fig. S26 and Fig. S27.

2. The authors prepared and compared MGC and OMGC. However, additional characterization about MGC and OMGC should be provided, for example, the electrical

conductivity of OMGC and the BET specific area of OMGC, and so on.

Response: Thank you for your constructive suggestion. The electrical conductivity of OMGC has been conducted, OMGC also presents high electrical conductivity of 235 S/m, which is two times higher than commercial activated carbon (10-100 S/m)¹⁶.

In addition, BET specific area of OMGC (1950 m²/g) has been obtained accompanied by N₂ adsorption/desorption isotherms in Fig. 1D.

To disclose the microstructure difference between MGC and OMGC, High resolution-TEM of OMGC was also provided (Fig. S13). The OMGC shows similar onion-like features with MGC, but shows local turbostratic graphitic microstructures.

The related comments have been added and marked with a yellow background in the text and Fig. S13.

3. The authors should explain in-detail all characterizations and figures, or present it in supporting information, for example, why did the authors give XPS results of different samples? In Ar, there are a lot of fluorine left over, while the XPS spectra of MGC and OMGC did not show F1s spectra. Why?

Response: Thank you for your valuable comments. We have carefully considered your comments, particularly your concern regarding the detailed analysis of related characterizations and figures. We have explained in-detail all characterizations and figures, particularly the figures in supporting information.

In terms of XPS, the MGC-products has a fluorine content of 15.4 at.% as displayed in **Table S2**. In fact, this sample is a lithiothermal reaction product without acid rinsing, which contains a large amount of lithium fluoride. There exists no obvious fluorine content in MGC and OMGC samples. In our revised manuscript, the explanation on “MGC-products” has been provided in the caption of Figure S22. We apologize for our oversight in this regard.

The XPS result confirms that there are only C and O elements in the surface of MGC and OMGC samples. For the sake of highlighting surface C and O configuration, well deconvoluted and fitted C 1s and O 1s XPS spectra are illustrated in Fig. S20-22. The three peaks of O 1s represent C=O groups, C–O groups, and O–C=O groups with

increasing binding energies, respectively. As shown in Fig. S20A and Fig. S21A, five peaks denoting sp^2 C, sp^3 C, C-O, C=O and O-C=O were deconvoluted with increasing binding energies. MGC and OMGC show 88 and 73 at.% of sp^2 C content, respectively, which confirms the highly graphitized structure for MGC and OMGC.

The related comments have been added and marked with a yellow background in the text, Fig. S2, Fig. S11, Fig. S16, Fig. S20, Fig. S21, Fig. S29, Fig. S31, Fig. S32 and Fig. S34.

4. For supercapacitors, the authors used with 1-ethyl-3-methyl-imidazolium tetrafluoroborate (EMIMBF₄) as the electrolyte. Why? I wonder if the MGC could show good electrochemical performance in an acidic or alkaline electrolyte.

Response: Thank you for your inquiry regarding the reason why 1-ethyl-3-methyl-imidazolium tetrafluoroborate (EMIMBF₄) as the electrolyte was used.

Ionic liquids are solvent-less electrolytes that possess excellent characteristics such as high thermal and chemical stability, negligible vapor pressure and broad electrochemical stability potential windows, making them potential candidates as electrolytes in energy storage devices (*Chem. Soc. Rev.*, 2015, 44, 7484-7539). And in terms of the wide potential window, ionic liquids are resistant to oxidation and reduction and many studies using ionic liquid-electrolyte-based supercapacitors could give operative cell voltages above 3 V, which can lead to improved energy densities. EMIMBF₄ has a relatively high ionic conductivity and gives an excellent overall performance among the common ionic liquids. Therefore, EMIMBF₄ was a good choice in this manuscript.

Despite narrow potential window for aqueous electrolytes, aqueous electrolytes have been used extensively in research and development due to their easy handling in the laboratory as compared to ionic liquids, which require purification procedures. Besides, aqueous electrolytes exhibit high conductivity as compared to ionic electrolytes, the electrochemical performances of MGC and thus were evaluated in a two-electrode system with 6 M KOH aqueous solution as electrolyte (Fig. R1).

Fig. R1A, B shows CV curves of MGC recorded at various scan rates from 1 mV

s^{-1} to 1000 mV s^{-1} . As can be seen, an increase in scan rate leads to the enlargement of current density apparently, meanwhile minor deviation and rectangular shape of satisfactory electric double-layer capacitive behavior for quick charge/discharge operations, suggesting that MGC shows good rate capability due to enhanced electric conductivity and high-speed diffusion channel.

The typical galvanostatic charge-discharge (GCD) curves at current densities ranging from 0.1 A g^{-1} to 50 A g^{-1} for MGC are displayed in Fig. R1C, D. As can be seen, all the GCD curves are nearly linear with an isosceles triangular shape, implying that sample MGC possesses excellent electrochemical reversibility and Coulombic efficiency. Gravimetric specific capacitance obtained from GCD curves at different current densities for MGC is shown in Fig. R1E. As can be seen, specific capacitances decrease gradually when elevating current density. For a current density of 0.1 A g^{-1} , the specific capacitance of MGC is 208 F g^{-1} . As the current density increased up to 50 A g^{-1} , the corresponding specific capacitance decreased to be 105 F g^{-1} , with a good rate capability of 50.5 %. Long-term cycling performance for MGC was examined at 10 A g^{-1} , presented in Fig. R1F. After 10000 cycles, relative high capacitance retention of 80.2 % can still be achieved. Additionally, our lab is trying to optimize the assembly process of supercapacitor in aqueous electrolytes to realize the full potential for MGC electrode.

Fig. R1. Electrochemical performances of MGC electrodes in a two-electrode symmetrical system in 6M KOH: CV curves tested at 1-40 mV s^{-1} (A), and 60-1000 mV s^{-1} (B); GCD curves tested at 0.1-2 A g^{-1} (C), and 4-50 A g^{-1} (D); Charge-discharge rate performance (E), and cycling performance at 10 A g^{-1} after 10000 cycles (F).

5. Check that all parts, insets, and Figures caption are mentioned, for example, Fig. S5. Check that the labels of the axes in all graphs are correct, for example, Fig. 3D.

Response: Thank you for your feedback. We appreciate your attention to detail and your commitment to improving the quality of our manuscript. In the revised manuscript, we have carefully checked and improved all parts of the images. Specifically, all parts, and figures caption have been mentioned for Fig. S5. And the labels of the axes in Fig. 3D have been checked.

The related comments have been added and marked with a yellow background in the text and Fig. S5 and Fig. 3D.

6. For comparison, the authors should give the CV curves, charge/discharge curves and long-cycle performance of OMGC.

Response: Thank you for your valuable feedback and suggestion. In the revised manuscript, we have provided the CV curves, charge/discharge curves and long-cycle performance of OMGC to clearly show the difference between MGC and OMGC.

Fig. S23 Supercapacitor performance of OMGC in EMIMBF₄ electrolyte. (A) CV curves from 1 to 100 V/s, which can sustain very high scan rates. (B) Galvanostatic charge/discharge curves at 0.1 to 200 A/g.

As can be seen from CV curves (Fig. S23A) for OMGC, an increase in scan rate leads to the enlargement of current density apparently, meanwhile a quasi-rectangular

shape could be maintained from 1 V s^{-1} to 50 V s^{-1} , while higher scan rate results in a significant distortion. A series of galvanostatic charge-discharge curves at current densities ranging from 0.1 A g^{-1} to 200 A g^{-1} for OMGC are displayed in Fig. S23B. OMGC shows obvious voltage drop especially at large discharge current density, leading to its slightly inferior rate capability than MGC.

After 100,000 charge/discharge cycles, the capacitance retention for OMGC is 89.2 %, which is competitive with the advanced carbon-based electrodes reported (Fig. 3F).

Fig. 3. Supercapacitor performance of MGC in EMIMBF₄ electrolyte. (A) CV curves from 1 to 100 V/s of MGC, which can sustain very high scan rates, like electrolytic capacitors. (B) Galvanostatic charge/discharge curves at 0.1 to 200 A/g for MGC. (C) Variation of specific capacitance with charge/discharge current density of MGC and OMGC. (D) Nyquist plots for MGC and OMGC capacitors taken at 3.5 V. The insets show magnification of the high frequency region. (E) Phase angle versus frequency for MGC and OMGC. (F) Capacitance retention for MGC and OMGC capacitors cycled at 10 A/g to 3.5V. (G) Variation of specific capacitance with charge/discharge current density of soft-package supercapacitor based on MGC and YP-50 in

EMIMBF₄. Their corresponding capacitance is 57 F and 52 F. Inset is the digital images of soft-package device with internal parallel structure.

The related comments have been added and marked with a yellow background in Fig. 3F and Fig. S23.

Reviewer #3 (Remarks to the Author):

The study presents development of Highly graphitized meso-microporous carbon materials by instantaneously gas-phase condensation that tends to form cross-coupled open channel under ultrahigh temperature, as electrode for supercapacitor application. the study is well supported by detail physiochemical and electrochemical characterizations. The study and results are significant to the field, and sufficient details have been provided. The manuscript can be considered for publication with minor revision.

Response: Thank you for your valuable feedback on our manuscript. We greatly appreciate your encouragement and commendation on our manuscript.

1. page 5, "Raman spectrum exhibits pronounced 2D peak (2690 cm⁻¹) of turbostratic graphite,". please note 2D peak is a measure of the stacking of layers in graphitic carbon. the statement must be corrected.

Response: Thank you for your instructive suggestion. The spectral shape of 2D peak is indicative for the degree and kind of stacking order of the nanosized graphene sheets which are formed during carbonization. It was noted from early studies that turbostratic graphite (i.e., without AB stacking) has a single 2D peak (*Carbon* 2020, 161, 359e372). However, its full width at half maximum (FWHM) is 50 cm⁻¹ almost double that of the 2D peak of graphene and up-shifted of 20 cm⁻¹. Here, the FWHM value of 2D peak for MGC is measured about 80 cm⁻¹, which is broad compared to the FWHM of turbostratic graphite. Besides, the 2D signal shape is symmetric and only one Lorentzian profile is needed for fitting the experimental data (Fig. S19), and the I_G/I_D ratio is greater than 1. All of these are indicative of highly graphitized carbon for MGC.

The related comments have been added and marked with a yellow background in the text.

2. Fig. 3 (B). please correct the caption of the X axis in GCD curve. Similar correction is also required in SI Fig. S24 (B).

Response: Thank you for your careful review. We appreciate your attention to detail and your commitment to improving the quality of our manuscript. The common caption of the X axis in GCD curve is time, while ultrashort time from ultrahigh current rate and long time from low current rate hardly show in one figure with all details identified. To show all details of curves at various current rates from 0.1A/g and 200A/g in one figure, we have adopted the same caption of the X axis in GCD curves with a related article, and carefully checked and improved all the GCD curves related (*Science* 2015, 350, 1508-1513.).

The related comments have been added and marked with a yellow background in Fig. 3B, Fig. S29B and Fig. S30D, E.

3. Page 14, Synthesis of MGC..."and clamp it using two Cu plates with 1 MPa.". please clarify the statement.

Response: Thank you for your kind review. PTFE powder (analytical grade) was enveloped into Li bag (3 cm×2 cm), and clamp it using two Cu plates with 1 MPa in a plate type thermocompressor.

The related comments have been added and marked with a yellow background in **Synthesis of MGC in Methods.**

4. There are a few grammatical errors, hence the manuscript must be thoroughly checked.

Response: Thank you for your valuable suggestion. I have checked the manuscript thoroughly, the grammatical errors have been corrected, and the English language in our manuscript has been polished as a whole by a native speaker.

REVIEWERS' COMMENTS

Reviewer #2 (Remarks to the Author):

The authors have fully addressed all my questions, and the manuscript can be accepted now.

Reviewer #3 (Remarks to the Author):

All the comments raised by the reviewers have been addressed in detail by the authors, and therefore the manuscript is now fit for publication.